# Widespread Distribution of Luteinizing Hormone/Choriogonadotropin Receptor in Human Juvenile Angiofibroma: Implications for a Sex-Specific Nasal Tumor

**DOI:** 10.3390/cells13141217

**Published:** 2024-07-19

**Authors:** Silke Wemmert, Martina Pyrski, Lukas Pillong, Maximilian Linxweiler, Frank Zufall, Trese Leinders-Zufall, Bernhard Schick

**Affiliations:** 1Department of Otorhinolaryngology, Head and Neck Surgery, Saarland University Medical Center, 66424 Homburg, Germany; silke.wemmert@uks.eu (S.W.); lukas.pillong@uks.eu (L.P.); maximilian.linxweiler@uks.eu (M.L.); 2Center for Integrative Physiology and Molecular Medicine (CIPMM), Saarland University, 66424 Homburg, Germany; martina.pyrski@uks.eu (M.P.); frank.zufall@uks.eu (F.Z.)

**Keywords:** LHCGR, LH, hCG, LH/FSH ratio, juvenile angiofibroma, nasopharyngeal, RNAscope, tumor, sex specific

## Abstract

Juvenile angiofibroma (JA) is a rare, sex-specific, and highly vascularized nasal tumor that almost exclusively affects male adolescents, but its etiology has been controversial. The G protein-coupled hormone receptor LHCGR [luteinizing hormone (LH)/choriogonadotropin (hCG) receptor] represents a promising new candidate for elucidating the underlying mechanisms of sex specificity, pubertal manifestation, and JA progression. We used highly sensitive RNAscope technology, together with immunohistochemistry, to investigate the cellular expression, localization, and distribution of LHCGR in tissue samples from JA patients. Our results provide evidence for LHCGR expression in subsets of cells throughout JA tissue sections, with the majority of LHCGR^+^ cells located in close vicinity to blood vessels, rendering them susceptible to endocrine LH/hCG signaling, but LHCGR^+^ cells were also detected in fibrocollagenous stroma. A majority of LHCGR^+^ cells located near the vascular lumen co-expressed the neural crest stem cell marker CD271. These results are intriguing as both LH and hCG are produced in a time- and sex-dependent manner, and are known to be capable of inducing cell proliferation and angiogenesis. Our results give rise to a new model that suggests endocrine mechanisms involving LHCGR and its ligands, together with autocrine and paracrine signaling, in JA vascularization and cell proliferation.

## 1. Introduction

Juvenile angiofibroma (JA, also known as JNA) is a rare, sex-specific, and highly vascularized tumor in the nasal cavity that almost exclusively affects male adolescents [1,2]. Patients usually suffer from unilateral nasal obstruction with or without severe nosebleeds, rhinorrhea, and reduced sense of smell. Although the tumor is benign, its rapid growth can severely damage the visual system and brain due to its space-occupying nature [3,4,5]. Stem cells positive for the neurotrophin receptor CD271 (NGFR/p75NTR) envelop the endothelial blood vessels in the fibrovascular architecture and are identified as cells of origin for JA [6]. Due to its sex specificity and pubertal manifestation, JA was originally thought to be an androgen-dependent tumor [7]. The spontaneous regression of JA after puberty supported this theory of hormonal influence [8]. However, investigations of sex hormone receptors and hormone blood levels for androgen, estrogen, and progesterone led to contradictory results [7,9,10,11] so that no general acceptance of a hormonal cause could be established. As an alternative receptor candidate, the luteinizing hormone/choriogonadotropin receptor (LHCGR) was previously identified in JA tissue by RT-PCR, but this has not attracted much attention since then [10]. 

LHCGR, a member of the superfamily of G protein-coupled receptors (GPCRs), mediates the action of both luteinizing hormone (LH) and human chorionic gonadotropins (hCG) [12,13]. Both LHCGR ligands are involved in the regulation of similar physiological functions in a time- and sex-specific manner, although hCG is more effective due to its higher receptor binding affinity and longer half-life. Placental hCG production stimulates testosterone secretion in male fetuses during early gestation, whereas the endogenous fetal hypothalamic–pituitary–gonadal (HPG) axis with its hypothalamic hormones LH and the follicle-stimulating hormone (FSH) begins to function in both sexes at mid-pregnancy [14,15]. A distinct sexual dimorphism can be observed in the ratio of LH to FSH. In boys, the LH level is higher than the FSH level, while girls have a higher ratio of FSH to LH [16,17,18,19]. The functional impact of hormonal activities in infancy is not fully resolved. LH and FSH levels decline again to lower levels until the start of a pulsatile GnRH (gonadotropin-releasing hormone) secretion marking the start of puberty. The pulsatility of the GnRH secretion is the force that triggers LH and FSH release, whereby LH levels during male puberty are again higher than FSH levels [17]. Due to the physiological changes in LH levels during adolescence, LHCGR expression could indicate the onset of tumor development and also explain the spontaneous regression of JA at the end of puberty, despite the fact that not all JA patients are affected in early childhood or puberty.

Besides its role in pregnancy, the glycoprotein hCG not only promotes the proliferation of stem cells, but can also be secreted by abnormal germ cells in benign or malignant tumors [20,21,22,23,24]. The hCG amino acid sequence supports five glycoproteins that can be produced by different cells having a wide range of functions depending on their glycosylation or sulfation states [25,26]. In addition, the hormone consists of two subunits, the alpha and beta subunits. The alpha subunit displays homologies with TSH, LH, and FSH, whereas the beta subunit has 85% homology with LH [27,28,29]. In pathological hCG production, tumors may produce disproportionately large amounts of either free alpha subunits or, more commonly, free beta subunits with various degrees of glycosylation. Aside from its endocrine role during pregnancy, hCG also has autocrine and paracrine functions [25,26,30]. The various hCG forms can bind to LHCGR with different affinities [31]. Since hCG is strongly involved in vascularization via LHCGR [32,33,34], the presence of both hCG and LHCGR could potentially provide positive hormonal feedback to induce tumor vascularization and cell proliferation leading to JA growth. Vascular endothelial growth factor (VEGF) is most commonly associated with an angiogenic effect, and is ubiquitously present in JA [35,36]. The relationship between VEGF and sexual dimorphism remains uncertain, and there is no clear evidence that it is related to pubertal status. Therefore, it is an inadequate explanation for the occurrence of JA in primarily adolescent males. hCG can however increase VEGF protein expression in a dose- and time-dependent manner [37] which, in turn, could promote angiogenesis via an MEK (mitogen-activated extracellular signal-regulated kinase)/ERK (extracellular signal-regulated kinase) signaling pathway [38]. Therefore, the activation of LHCGR could act as a trigger for angiogenesis in this benign sex-specific tumor.

On the basis of the previous results, we hypothesized that LHCGR-expressing (LHCGR^+^) cells could be located in close proximity to the CD271^+^ stem cells which are found near the individual vasculature of JA tissue. In this study, we aimed to identify the exact cellular expression, distribution, and localization of LHCGR in intact cells of tissue samples obtained from JA patients. We visualized *LHCGR* RNA using the RNAscope platform of ultrasensitive single-molecule fluorescence in situ RNA hybridization and also performed immunohistochemistry to visualize LHCGR protein with cellular resolution. Our results provide new strategies to gain insight into the hormonal dependence of JA development and proliferation and indicate that LHCGR may be a valid candidate for therapeutic intervention. 

## 2. Materials and Methods

### 2.1. Tumor Specimens

Formalin-fixed, paraffin-embedded tissue samples (FFPE) from 10 male JA patients aged 11 to 20 years who underwent surgery and diagnosis at Saarland University Hospital between 2010 and 2023 were analyzed. Written informed consent was obtained from all patients and the use of human tissue was performed according to the Code of Ethics of the World Medical Association (Declaration of Helsinki) and approved by the Institutional Review Board (#218/10) of Saarland University.

### 2.2. RNAscope Fluorescence In Situ Hybridization

Paraffin sections (8–10 µm) of 5 patients were collected and subjected to RNAscope in situ hybridization using the RNAscope^®^ Multiplex Fluorescent Reagent Kit v2 Kit (Cat. No. 323100, ACD Bio-Techne, Newark, CA, USA) and a probe specific for the human LHCGR (hs-LHCGR, Cat. No. 435531) according to the manufacturer’s protocol. Briefly, sections were deparaffinized for 1 h at 60 °C, washed twice for 8 min in xylene, dehydrated in 100% ethanol (2 × 4 min), and air-dried. Sections were then rehydrated in phosphate-buffered saline (PBS, pH 7.2) for 10 min, incubated in RNAscope^®^ H_2_O_2_ for 10 min, rinsed twice in distilled water, incubated in Target Retrieval Reagent (Cat. No. 322000, ACD Biotechne, Newark, CA, USA) for 15 min at 95 °C, and rinsed twice in distilled water prior to RNAscope^®^ Protease Plus treatment for 30 min at 40 °C. After two rinses in distilled water, sections were hybridized with the *hsLHCGR* probe, which was designed as a channel 1 probe containing 20 ZZ structures targeting base pairs 940-1991 of NM_000233.4 (NIH GenBank, NCBI Reference Sequence), specific for exons 9 and two thirds of exon 10. Negative and positive RNAscope controls were performed (Appendix A). As a negative control, we used a channel 1 probe for the *dapB* gene of Bacillus subtilis strain SMY provided by the manufacturer (Cat#320871, ACD Bio-Techne, Newark, CA, USA). As positive control, we used ovarian tissue obtained from an ovarian carcinoma patient (#2346-1) that expresses *LHCGR*. This tissue was kindly provided by Prof. Dr. R. M. Bohle, Institute of Pathology, University Hospital of Saarland, Germany. After hybridization (2 h, 40 °C), sections were rinsed twice for 2 min in wash buffer (ACD Bio-Techne), sequentially treated with v2-amplification reagents at 40 °C (30 min Amp1, 30 min Amp2, 15 min Amp3, and 15 min HRP-C1), and incubated in tyramide-amplification-based fluorescence reagent (15 min, 40 °C TSA Vivid Dye 650, Tocris Cat. No. 5748, Bristol, UK) with intermittent washes at room temperature between reagents. Cell nuclei were stained with DAPI solution (ACD Bio-Techne) for 1 min, then tissue sections were coverslipped in fluorescence medium (DAKO, Glostrup Kommune, Denmark) and stored at 4 °C for at least 12 h. Fluorescence images were acquired as 1 µm optical sections using an upright scanning confocal microscope (Zeiss LSM 880 Indimo, Oberkochen, Germany) equipped with a 32-channel GaAsP-PMT, 2-channel PMT QUASAR detector using C-Apochromat objectives (Zeiss, Oberkochen, Germany) 10×/0.45 W, 40×/1.2 W Korr UV–VIS–IR or 63×/1.2 W Korr UV–VIS–IR [39,40,41]. All scanning head settings were kept constant. The RNAscope probe was visualized by excitation at 647 nm (emission 638–747 nm). DAPI nuclear stain was visualized by excitation at 355 nm (emission 401–504 nm). Higher magnification confocal images are Z-stacks presented as maximum intensity projections consisting of 10–16 confocal sections, each 0.4 µm in thickness. The digitized images exported from ZEN software (Version Black v.14.0.25.201, Zeiss, Oberkochen, Germany) were assembled, and minimally adjusted in contrast and brightness using Photoshop Elements 10 (Adobe Photoshop, San Jose, CA, USA).

### 2.3. Quantification of RNAscope Data

To determine the number of cells positive for *LHCGR* obtained by RNAscope, high-resolution 10× images (4416 px × 4416 px; 850 µm × 850 µm) were captured using the Zeiss LSM 880 confocal microscope (see above) and analyzed with Fiji/ImageJ 1.53 software (NIH, Bethesda, NY, USA). The vessel lumen was outlined using the freehand tool, and the circumference of 50 µm was automatically calculated using the ‘enlarge’ function. LHCGR^+^ cells were manually counted using the ‘multipoint tool’ of Fiji/ImageJ 1.53 software. The presence of a nucleus and a minimum of three small puncta corresponding to labeled RNA molecules were verified for each positive counted cell. The number of cells per mm^2^ within a given region of interest was calculated. 

### 2.4. Immunohistochemistry

Formalin-fixed, paraffin-embedded sections of 9 patients were collected and subjected to LHCGR and CD271 immunohistochemistry. From each sample, 3 µm tissue sections (Leica RM 2235 microtome, Leica Microsystems, Wetzlar, Germany) were collected on Superfrost Ultra Plus glass slides (Menzel, Bielefeld, Germany) and dried overnight at 37 °C. Following deparaffinization and rehydration, sections were subjected to heat-induced antigen retrieval under acidic conditions (10 mM sodium citrate, pH 6.0) for 15 min at ≥95 °C using a conventional rice cooker (Tefal Classic 2, RK1011, Tefal, Berlin, Germany). After cooling for 30 min to RT (20 °C), sections were washed three times for 3 min in PBS. Prior to antibody application, unspecific protein binding capacities were blocked by incubating sections for 30 min in 5% BSA prepared in PBS. For the co-detection of CD271 and LHCGR antigens, primary antibodies were applied sequentially. Sections were first incubated overnight with rabbit anti-CD271/NGFR monoclonal antibody (1:1000, MA5-31968, Invitrogen, Darmstadt, Germany; RRID: AB_2809262) diluted in PBS containing 1% BSA. Bound antibody was visualized using the Dako REALTM detection system protocol (K5005, Agilent, Waldbronn, Germany), equipped with a biotinylated anti-rabbit secondary antibody, a streptavidin-conjugated alkaline phosphatase, and the Fast Red chromogenic substrate yielding a red color precipitate. Subsequently, sections were blocked with 5% BSA prepared in PBS for 30 min and then incubated overnight with anti-human LHCGR mouse monoclonal antibody (1:500, ab218907, Abcam, Cambridge, UK; RRID: AB_3096979) diluted in PBS containing 1% BSA. Bound LHCGR antibody was visualized using the Dako REALTM detection system protocol (K5007, Agilent, Waldbronn, Germany), equipped with a horse-radish peroxidase (HRP)-conjugated anti-mouse secondary antibody and the chromogenic substrate DAB (3,3′-Diaminobenzidine), yielding a brownish color precipitate. Finally, nuclei were counterstained with Mayer’s Hematoxylin (MHS32, Merck, Darmstadt, Germany). All steps were carried out at RT; incubation of primary antibodies was at 4 °C.

As a negative control, we applied antibody dilution buffer without primary antibodies (Appendix A). As a positive control, we used ovarian tissue obtained from an ovarian carcinoma patient (patient #2346-1) which expresses LHCGR and CD271 (Appendix A), kindly provided by Prof. Dr. R. M. Bohle, Institute of Pathology, Saarland University Medical Center. Bright field images were captured at 20× magnification on an BX61 microscope attached to a DP71 camera (Olympus, Hamburg, Germany).

### 2.5. Quantification of Immunohistochemistry Data

We employed unbiased, computer-assisted analyses to quantify the overlap between LHCGR^+^ and CD271^+^ areas. The color threshold tool of ImageJ 1.53 (NIH, Bethesda, MD, USA) separated the detection of the two proteins. In order to detect LHCGR immunoreactivity, a threshold method based on the Otsu algorithm and the HSB color space was employed with a hue setting of 19 to 120. This prevented the detection of the purple nuclear staining and the more magenta CD271 staining. To detect CD271 immunoreactivity, the HSB color space was set with a hue between 204 and 255 preventing the detection of the purple nuclear and the brown LHCGR staining. By measuring the total field of view (FOV) area and the areas containing either LHCGR^+^ cells or CD271^+^ cells or both, we determined the percentage of LHCGR^+^ area within CD271^+^ or CD271^−^ regions. Depending on the size of the surgically removed JA tissue, 6–9 representative FOVs from the tissue section of each patient (n = 9), captured with a 20× oil immersion objective (Olympus UPlanSApo 20×/0.85, Hamburg, Germany), were examined.

The total number of LHCGR^+^, LHCGR^+^ CD271^+^, or CD271^+^ cells per mm^2^ tissue was manually counted using the ‘multipoint tool’ of the Fiji/ImageJ 1.54 software. In analogy to the RNAscope analyses, the number of cells located in the 50 µm area surrounding the lumen of the blood vessel was analyzed. Two independent observers both unaware of the results from the computer-assisted analysis examined 6–16 representative FOVs in a given tissue section of each patient (n = 9) captured with a 20× or 40× oil immersion objective (Olympus UPlanSApo 20×/0.85; UApo/340 40×/1.35, Hamburg, Germany), were examined. We used the same criteria with respect to the number of cells located in a 50 µm area surrounding the border of a blood vessel as in the RNAscope analyses.

### 2.6. Statistics

Statistical analyses were performed using Origin Pro (OriginLab Corporation, Northampton, MA, USA). Assumptions of normality and homogeneity of variance (Kolmogorov–Smirnov, Shapiro–Wilk) were tested before conducting the following statistical approaches. Student’s *t*-test was used to measure the significance of the differences between two distributions. Multiple groups were compared using a one-way analysis of variance (ANOVA) with the LSD multiple comparison test as a post hoc comparison. In case the results failed the test of normality, multiple groups were compared using the Kruskal–Wallis ANOVA with Dunn’s least test as a post hoc comparison. The probability of error level (alpha) was chosen to be 0.05. Unless otherwise stated, data are expressed as mean ± SD. Box plots display the interquartile (25–75%) ranges, median (line), and mean (black square) values with whiskers indicating SD values. 

## 3. Results

### 3.1. LHCGR RNA Exists in Individual Cells from JA Tissue

mRNA for the hormone receptor *LHCGR* was previously identified in JA tissue by RT-PCR [10] and could represent a new candidate to investigate the hypothesis of hormonal imbalance in this benign fibrovascular tumor in the nasal cavity. To enable cellular mapping of *LHCGR* in JA tissue sections, we opted for RNAscope in situ hybridization technology, which visualizes a single RNA molecule as a dot or punctum [42,43] using highly specific RNAscope probes designed to recognize a portion of the *LHCGR* coding sequence (see Section 2). RNAscope was performed on JA tissue sections from 5 male patients between the age of 13–18 years (Figure 1, Figure 2 and Figure 3). In all cases, we found that *LHCGR* was mostly enriched in cells located around blood vessels, identified by DAPI nuclear staining outlining the structure of individual blood vessels (Figure 1A–C). Representative examples from five JA patients are shown in Figure 2A–O. Higher magnification of the vascular region clearly shows the heterogeneity of *LHCGR* staining within individual tissue sections. Some blood vessels show sparse labeling of cells with few or no *LHCGR* puncta, while prominently labeled blood vessels display cells with densely packed mRNA puncta forming broader clusters as exemplified in patient #1180 (Figure 2E). The patterns of puncta in individual cells from patient #2420 are noticeably different from those of JA patients with fully mature angiofibroma, ranging from some few nuclear and subcellular *LHCGR* puncta to large labeled regions that cover most of the DAPI-stained nuclei (Figure 2M–O). Such extensively labeled *LHCGR* regions are likely due to the great number of fluorescent puncta obscuring the identification of individual puncta. Consistent with this particular staining pattern, the surgeon classified this patient #2420 as in remission at the time of the tissue removal. *LHCGR^+^* cells were mainly found in the vicinity of blood vessels, with very few vessels showing no labeled cells (Figure 3A, *p* < 0.022). The number of blood vessels in JA tissue sections treated and analyzed with RNAscope followed a pattern that suggested a dependence on development (13-year-old patient) and end of puberty (18-year-old patient in remission) (Figure 3B). To quantify the presence of *LHCGR^+^* cells near the vasculature, all nuclei containing at least three individual *LHCGR* puncta were counted within a 50 µm area surrounding the border of the vascular lumen (Figure 3C–H). In all patients, we observed *LHCGR^+^* cells in the vicinity of blood vessels (Figure 2H). However, tissue from patient #2420 was significantly different from that of the other JA patients. This sample had the lowest number of *LHCGR^+^* cells (*p* < 0.001) together with a low number of blood vessels, whereby 3 of the 15 blood vessels showed no *LHCGR* labeling in the 50 μm vicinity of the vascular border.

Overall, the RNAscope results provided conclusive evidence for the presence of *LHCGR* expression in subsets of individual cells from JA tissue, with the majority of JA vessels exhibiting *LHCGR^+^* cells in their close vicinity. 

### 3.2. LHCGR Protein Is Mainly Located near the Vasculature

Given that the neural crest stem cell marker CD271 was specifically detected around the pathological vasculature of JA patients [6], we examined the presence and co-localization of cells expressing LHCGR protein (LHCGR^+^) in CD271^+^ and CD271^−^ areas of JA tissue using unbiased computer-assisted analyses. Immunohistochemistry for LHCGR and CD271 was performed on tissue sections from nine JA patients aged between 11 and 20 years, whereby four of the nine JA patients were initially analyzed using RNAscope. Examples of the JA histology are shown in Figure 4, with one patient having a recurrence of JA (Figure 4D) and one patient being in remission (Figure 4H). The recurrence of this benign tumor is assumed to be due to residual tumor tissue that could not be removed during surgery because of its location. In all surgically removed JA tissue, LHCGR^+^ cells were observed either in close vicinity of blood vessels or within fibrocollagenous stroma. Those blood vessels with LHCGR^+^ labeling in their vicinity often showed structural abnormalities, typical for the highly vascularized JA regions [44]. CD271^+^ labeling surrounded most of these vascular regions (Figure 4). An exception was patient #2420 where no CD271^+^ regions could be found (Figure 4H), suggesting that a loss of the stem cell marker could indicate a JA remission. Furthermore, the vascular architecture of the youngest patient (Figure 4A) clearly stands out from the other examples (Figure 4B–G,I). Several small, strongly stained CD271^+^ areas are observed next to LHCGR^+^ regions near the vessels, with occasional light CD271^+^ staining, likely indicating the presence of a not yet fully developed JA in this young patient, who is probably at the beginning of puberty. Co-localization analyses confirmed that LHCGR was mainly present in the vascular-enriched CD271^+^ regions rather than in CD271^−^ regions (Figure 5A; *p* = 0.012). Overall, 6–9 representative fields of view (FOVs) in a given JA tissue slice from each patient (n = 9) were examined and documented (Figure 5B,C; see Section 2). 

We observed LHCGR^+^ cells in both CD271^+^ and CD271^−^ regions from samples in all JA patients, although with quantitative differences. An exception in the presence of LHCGR^+^ cells in the CD271^+^ area was patient #2420, who was found to be in remission (Figure 5B). This patient lacked any CD271^+^ staining (Figure 5B). The youngest patient, #1041 (11 years old), also stood out in this analysis due to the low co-localization of areas that were both LHCGR^+^ and CD271^+^. The low number is due to the fact that only a few areas were CD271^+^ and most of the vascular regions were expressing only LHCGR protein but not CD271. The heterogeneity in the percentage of LHCGR^+^ regions in CD271^+^ areas is also apparent in patients with a fully developed JA, albeit with much higher variability and more increased presence of LHCGR^+^ areas colocalizing with CD271 (Figure 5B). Our quantitative analyses suggest that the extent of co-localization of LHCGR and CD271 staining could indicate the presence of a fully mature JA (Figure 5B).

LHCGR was also identified in CD271^−^ areas in samples from all JA patients (Figure 5C). The highest level of LHCGR immunoreactivity without CD271 labeling was observed in patient #1562 (*p* < 0.001). This patient exhibited a recurrence of JA. Given the anatomical constraints of surgical resection, complete removal of JA may fail. Consequently, recurrence represents a prevalent phenomenon here which can be defined as the persistence of residual JA tissue. Stem cells are the reactive cells that trigger cell proliferation, regardless of whether the nasal cavity of healthy patients contains LHCGR^+^ cells. The recurrence of JA may be attributed to the presence of CD271^+^ cells in proximity to small vessels, which could render LHCGR^+^ cells susceptible to endocrine stimulation, thereby contributing to the recurrence of JA. LHCGR^+^ cells seem to persist longer than CD271^+^ cells, as observed in the remission patient #2420 (Figure 4H). We also note that the sample of recurrent patient #1562 showed extended areas of blood vessels containing LHCGR^+^ cells, as well as areas of small vascular spaces that contained cells expressing both LHCGR and CD271 (Figure 5D). Further studies with a larger sample size from patients showing a recurrence will be needed to substantiate this point. 

### 3.3. The Presence of LHCGR in CD271^+^ Cells Is Indicative of a Fully Mature JA

LHCGR and its ligand hCG are known to play important roles in angiogenic and metastatic processes [24,32,33] and certain stem cells can secrete hCG [20]. Therefore, the co-existence of stem cells and LHCGR^+^ cells in JA tissue could result in positive hormonal feedback inducing tumor vascularization and cell proliferation, thus ultimately leading to JA growth. To explore this idea further and to validate the findings from the computer-assisted analyses, we undertook an in-depth analysis of the distribution and cellular composition of LHCGR^+^ and CD271^+^ cells in proximity to the blood vessels in 6–16 FOVs of a given JA tissue section from each patient (n = 9, Figure 6A). Quantitative analyses of the number of cells/mm^2^ in the samples revealed a greater number of LHCGR^+^ cells than CD271^+^ cells (*p* < 0.001; Figure 6B), whereby the majority of LHCGR^+^ cells were not CD271^+^ stem cells (*p* = 0.294). These findings indicate that a significant proportion of LHCGR^+^ cells are present in the fibrocollagenous stroma and lack CD271 expression. However, cells expressing only LHCGR can also be observed in close proximity to the vessels (Figure 4 and Figure 5D). In contrast, the majority of CD271^+^ cells simultaneously express LHCGR (*p* = 0.525), which would render the cells in close proximity to the blood vessel wall susceptible to endocrine signaling by fluctuations in serum LH levels.

The number of blood vessels exhibited considerable variability among all samples (Figure 6C). The average number of blood vessels counted in samples of the 9 JA patients was 56 ± 37 per FOV, with a range from 11 to 135 vessels/FOV. The lowest number of blood vessels was observed in patient #2420 who was in remission. Together with the RNAscope data (Figure 3B), this result confirms previous findings of robust formation of blood vessels in JA tissue, which appear to be susceptible to regression under conditions that could depend on reduced LH levels at the end of puberty or, alternatively, on hCG-secreting stem cells.

To explore in greater detail the cellular expression of LHCGR and CD271 in proximity to the blood vessel walls, a 50 μm area surrounding the border of each vascular lumen was examined in samples from all patients (Figure 6D–F). LHCGR^+^ CD271^−^ cells were unequivocally detected in all JA samples, despite some variations (Figure 6D). The presence of double-labeled cells expressing both LHCGR and CD271 was observed in the majority of cases, with the exception of patients #1041 and #2420 (Figure 6E). This finding is consistent with the unbiased computer-assisted analysis presented in Figure 5. Patient #1041 (11 years old) exhibited a low number of double-labeled cells (Figure 6E), which could indicate an early detection of a developing angiofibroma. A cell density of LHCGR^+^ CD271^+^ cells > 5 cells/mm^2^ (median of the lowest 1st quartile of patient #1041) seemed to indicate a fully developed tumor. By contrast, the JA remission patient #2420 showed no double-labeled cells; CD271^+^ cells were absent (Figure 6E,F). In all other cases, the samples contained CD271^+^ cells near the vasculature (Figure 6F).

Collectively, these results provide evidence for the co-existence of CD271^+^ stem cells and LHCGR^+^ cells in the vicinity of the JA vasculature. The stem cells themselves can also express LHCGR, and their presence is indicative of a fully developed JA. These results suggest the possibility for autocrine as well as paracrine signaling underlying JA progression, in addition to endocrine mechanisms involving LHCGR. 

## 4. Discussion

A number of hormonal imbalances have been proposed to explain the strong predisposition for a fibrovascular tumor–juvenile angiofibroma–in the nose of young male adolescents. General acceptance of a hormonal imbalance could not be established thus far because of the considerable variability in age at diagnosis and the rare involvement of very young patients who do not conform to typical male pubertal growth patterns. The previously identified mRNA of the G protein-coupled receptor LHCGR in JA tissue [10] would suggest a highly intriguing explanation for sex specificity, pubertal manifestation, and JA progression. LHCGR not only binds LH, which plays a role during minipuberty in infancy and puberty in adolescence [16], but also hCG, which is important during pregnancy and plays a role in various tumors [24,32,33]. Sexual dimorphism is evident in the LH/FSH ratio, which is higher in boys than in girls during gestation, minipuberty, and prepuberty [16,17,18,19]. To further strengthen a potential role of LHCGR in JA, it was necessary to provide clear evidence for its expression and localization in subsets of individual cells from JA tissue samples. For example, a proposed endocrine involvement of LHCGR would likely require that the receptor be located in close proximity to the JA vasculature.

Here, we have provided several key findings that significantly advance our understanding of this concept: (1) As predicted by the previous low levels of *LHCGR* mRNA in JA [10], we obtained profound evidence for cellular *LHCGR* expression using RNAscope technology and showed receptor localization in numerous, isolated cells throughout JA sections, with the majority of *LHCGR*^+^ cells located near the vasculature. (2) Consistent with these results, LHCGR immunoreactivity was predominantly present in vascular regions, but LHCGR^+^ cells were also detected in fibrocollagenous stroma. (3) The majority of those LHCGR^+^ cells that were localized exclusively in the vicinity of the vascular lumen also co-expressed the stem cell marker CD271. These results suggest that cells in close proximity to the blood vessel wall may indeed be susceptible to endocrine signaling by fluctuations in serum LH levels, as depicted in the model of Figure 7. (4) The co-expression of LHCGR and CD271 in double-labeled cells is indicative of a fully mature JA, whereby low co-expression could indicate either a developing JA or a JA in remission. (5) The proximity of double-labeled LHCGR^+^ CD271^+^ cells close to other LHCGR^+^ CD271^−^ cells may indicate the existence of autocrine and paracrine signaling mechanisms in addition to endocrine effects in JA tissues (Figure 7).

Errant stem cells have the potential to induce tumors. CD271^+^ neural crest stem cells are proposed as the origin of JA [6]. CD271 is a neurotrophin receptor and a member of the tumor necrosis factor receptor superfamily which functions as either a pro- or anti-tumorigenic effect. Interestingly, some neural crest cells retain their capacity for multipotent differentiation, can self-renew or simply remain quiescent after migration, effects that can persist even into adult life [45]. The mechanisms underlying induction and differentiation of neural crest cells into distinct cell types remain incompletely understood. Yet, the association or combination of a multipotent stem and progenitor cell type can contribute to normal as well as abnormal vascular development [46]. In this study, we found that the majority of CD271^+^ cells also express LHCGR. In female reproduction, activation of this hormone receptor not only stimulates ovulation, but also angiogenesis in the ovaries. Both LH and hCG are thus capable of inducing cell proliferation whereby hCG seems to be the more angiogenic factor [34,47].

LHCGR has recently been considered as a prognostic marker for a subset of tumors, including the growth of testicular germ cell tumors (seminomas) [48]. Furthermore, LHCGR activity has been implicated in the induction and progression of some cancers [25,28]. Patients with a malignant endometrial adenocarcinoma have been found to benefit from treatments with inhibitors of either LHCGR or the vascular endothelial growth factor receptor 2 (VEGFR2) [49]. VEGF is a well-known angiogenic factor and a primary stimulant of tumor vascularization and it is widely present in JA [35,36]. However, VEGF is not directly related to pubertal status and is largely not sex-specific, rendering it an inadequate explanation for the occurrence of JA in primarily adolescent males. Importantly, hCG can upregulate VEGF protein expression in a dose- and time-dependent manner [37] which, in turn, could promote angiogenesis via a MEK/ERK signaling pathway [38]. hCG could thus potentially act as a trigger for angiogenesis in this benign sex-specific tumor.

The two hormones LH and hCG generally are produced in a time- and sex-dependent manner and are known to have highly specific physiological functions. It is assumed that they fulfill different roles in the regulation of fetal and neonatal functions, many of which have not yet been conclusively clarified [13,16]. During male development, LH is regularly present and could be a trigger for the development of neovascularization and cell proliferation, which would explain the normally high prevalence of JA during male adolescence. As serum LH concentrations in girls are much higher, especially during gestation, LH could alternatively protect girls from developing JA. On the other hand, the LH/FSH ratio is higher in boys. This LH/FSH ratio is an important clinical tool in the diagnosis of female reproductive diseases, e.g., precocious puberty or polycystic ovary syndrome [50,51]. In JA, it could also indicate a complex interaction between LHCGR and FSHR activity. *FSHR* RNA is reportedly upregulated in JA tissue [10]. 

Currently, the most likely trigger for CD271^+^ stem cells to initiate cell proliferation and angiogenesis is an elevated LH level during adolescence (Figure 7). The LH level is approximately 3–4 times higher at the onset of male puberty compared to gestation and prepubertal development in boys [16]. If the CD271^+^ cells in JA tissue are able to secrete hCG, the simultaneous presence of LH and hCG could promote a proliferation of cell activity and the formation of new capillaries. hCG is secreted by certain stem cells as well as cells that have been obtained mainly from female benign or malignant tumors [20,21,22,23,24]. The level of hCG in male tumors is generally not considered a target in the various clinical disciplines as it is still assumed that hCG is not present in males, despite the fact that hCG levels have been reported to be elevated in males in a number of malignant tumors [25,26,52].

## 5. Conclusions

In conclusion, the presence of LHCGR in cells of JA tissue could lead to positive hormonal feedback onto stem cells via endocrine, paracrine, or autocrine mechanisms (or combinations of these, Figure 7). These effects could induce vascularization and cell proliferation of the tumor and ultimately lead to a benign nasal angiofibroma. Several questions remain unanswered. The initial trigger causing the activation of CD271^+^ cells still remains unclear. In addition to an adequate LH concentration to trigger JA at the onset of puberty, the LH/FSH ratio and thus the presence of FSHR should be determined in future experiments. Furthermore, it has not been proven yet that the CD271^+^ stem cells in JA tissue can secrete hCG. Consequently, it is necessary to assess LH, hCG, and FSH levels in serum and urine samples from patients of all ages, and potentially even in JA tissue, in order to substantiate LHCGR as the target and cause of hormonal dependence in JA development and proliferation. Due to the limited availability of JA patients in our clinic, the aforementioned hormone levels could not yet be determined for our research purposes. Furthermore, the use of sophisticated and rigorous cell marker panels, including precise genetic tools for lineage labeling and assays for vascular stem and progenitor cells, should enable the determination of the subset of (stem) cells involved. A more comprehensive understanding of LHCGR^+^ CD271^+^ cells and LHCGR^+^ CD271^−^ cells in the vicinity of the vasculature could facilitate the development of more effective therapeutic strategies or even prompt early intervention in JA.

## Figures and Tables

**Figure 1 cells-13-01217-f001:**
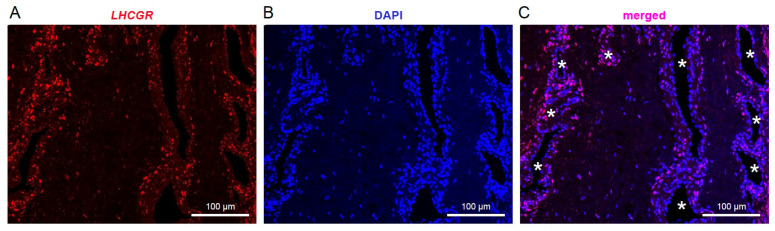
*LHCGR* is enriched in cells located around blood vessels. Examples show confocal fluorescence images of RNAscope in situ hybridization in JA tissue samples using a probe specifically directed against *LHCGR*. (**A**–**C**) *LHCGR* expression (red) is primarily restricted to cells surrounding the blood vessels but also scattered in the tissue between blood vessels (**A**,**C**). In addition to the empty dark vascular lumen, blood vessels can be identified by DAPI (4′,6-diamidino-2-phenylindole, blue) nuclear staining outlining their structure (**B**). An overlay of the two images (merged, **C**) summarizes the data of patient #1754 (17 y). Blood vessels are marked with an asterisk.

**Figure 2 cells-13-01217-f002:**
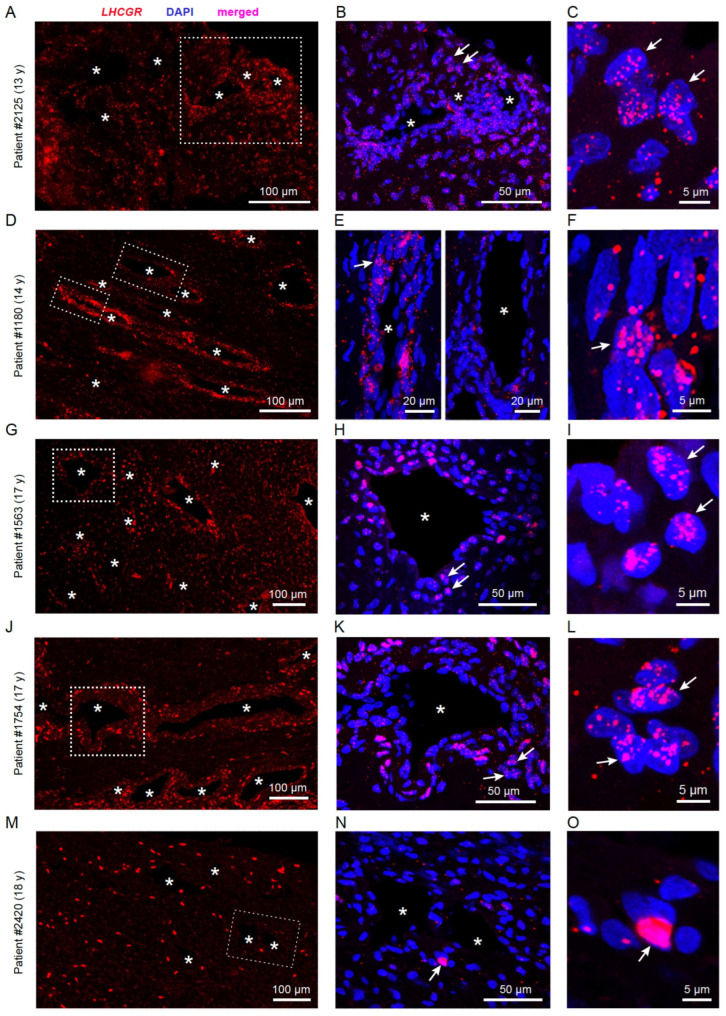
(**A**–**O**) *LHCGR* RNAscope performed on JA tissue sections from 5 patients. Confocal fluorescence images of JA tissue from three patients showing representative views of the widespread *LHCGR* distribution (**A**,**D**,**G**,**J**,**M**). The dashed white boxes are shown in (**B**,**E**,**H**,**K**,**N**) as merged images of *LHCGR* expression and DAPI labeling from the vascular regions. White arrows indicate the nuclei magnified in (**C**,**F**,**I**,**L**,**O**) showing the distribution of RNA puncta above and next to the nuclei. Blood vessels are marked with an asterisk.

**Figure 3 cells-13-01217-f003:**
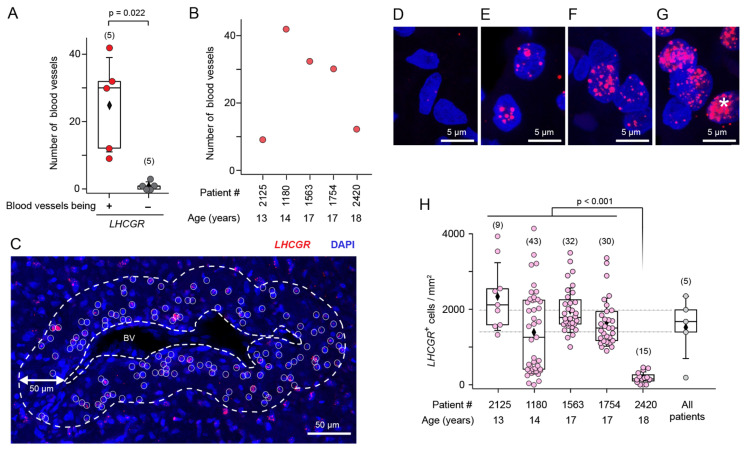
Analyses of *LHCGR* expression in cells surrounding the JA vasculature. (**A**) JA tissue contains more blood vessels with *LHCGR*^+^ cells in its surroundings compared to blood vessels without *LHCGR* expression (paired *t*-test: t(4) = 3.736, *p* < 0.020). (**B**) Distribution of the number of blood vessels showing *LHCGR* expression in nearby cells in each patient, sorted by age. (**C**–**G**) Assessment of *LHCGR*^+^ cells surrounding the blood vessels. (**C**) The boundary of the blood vessel lumen (BV) and its 50 μm circumference (dashed lines) indicate the region within which *LHCGR* puncta (*red*) were counted per nucleus (DAPI, *blue*), indicating the presence of a given cell. (**D**) Nuclei with >3 puncta were counted as *LHCGR*^−^. (**E**–**G**) Nuclei with >3 puncta were counted as *LHCGR*^+^. Number and size of *LHCGR* puncta varied from cell to cell. Especially large and occasionally diffusely stained *LHCGR*^+^ regions in a given cell are due to the large number of fluorescent puncta (asterisk). (**H**) Boxplot of the number of *LHCGR*^+^ cells/mm^2^ near blood vessels in tissue sections from each individual patient sorted by age (*pink* puncta) and summarized for all patients (*grey* puncta). Numbers in parentheses indicate the total number of blood vessels analyzed for a given patient (*pink*) or the number of patients in the case of the summary boxplot (*grey*). Kruskal–Wallis ANOVA: χ^2^(4) = 43.84, *p* < 0.0001, post hoc: Dunn’s.

**Figure 4 cells-13-01217-f004:**
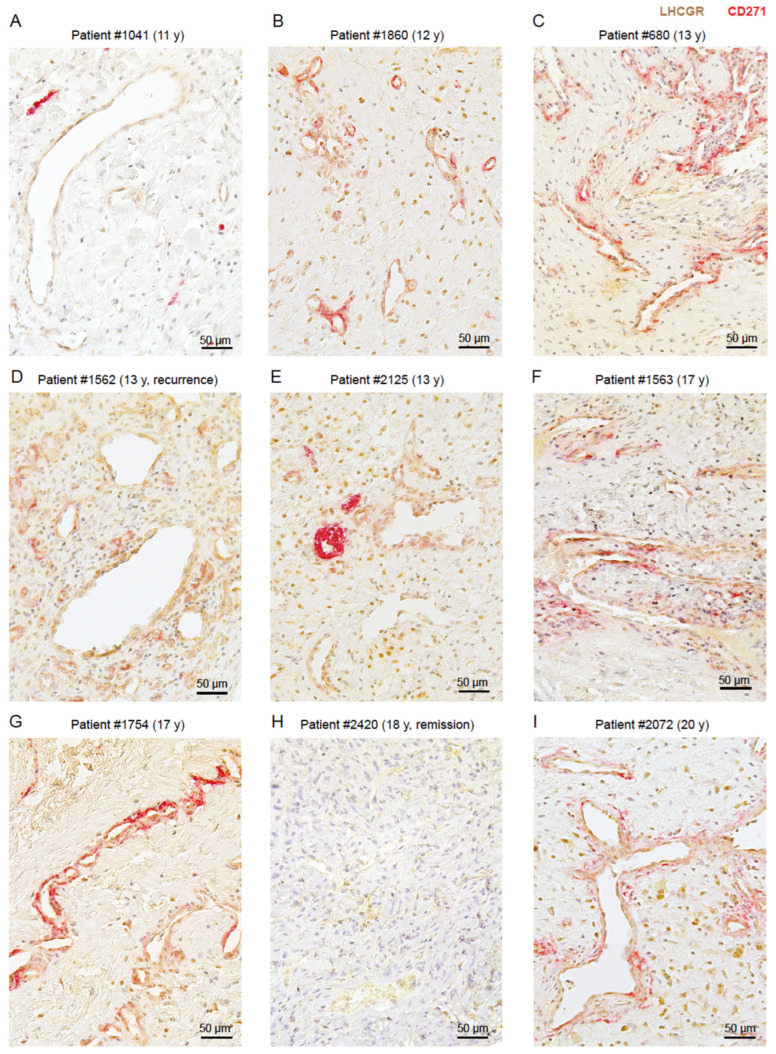
LHCGR protein is in stroma and near the vasculature of JA tissue. (**A**–**I**) Examples show the heterogeneous immunoreactivity for LHCGR (*brown*) and CD271 (*red*) performed on tissue sections from 9 JA patients aged between 11 and 20 years. Sections were stained with specific antibodies and developed using enzymatic staining (see Section 2). Nuclei are stained using hematoxylin.

**Figure 5 cells-13-01217-f005:**
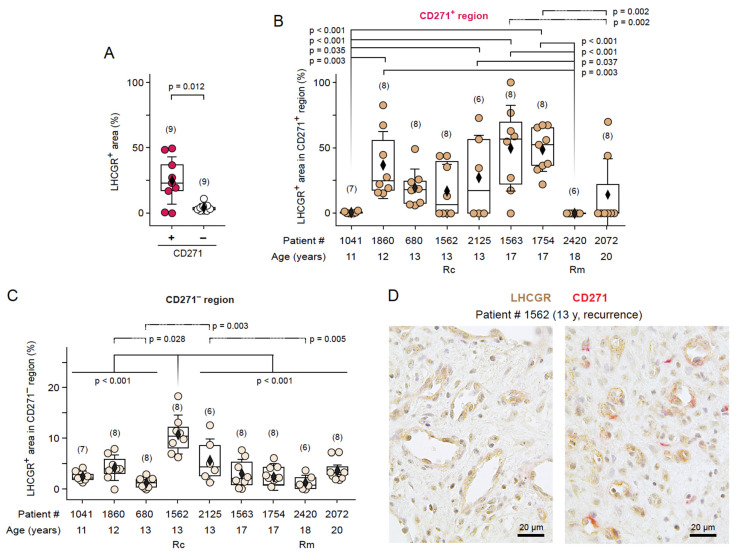
LHCGR immunoreactivity is mainly detected near the vasculature as revealed by computer-assisted analysis. (**A**) Co-localization analyses of LHCGR and CD271 expression in 9 JA patients indicate that LHCGR was mainly present in the vascular-enriched CD271^+^ regions rather than in CD271^−^ regions (*t*-test with Welch correction, equal variance not assumed: t(8.397) = 3.184, *p* = 0.012). (**B**) Box plots of the percentage of LHCGR^+^ area within CD271^+^ regions per patient, sorted by age; ANOVA: F(59) = 5.230, *p* < 0.0001; post hoc: LSD. (**C**) Box plots of the percentage of LHCGR^+^ area in CD271^−^ regions per patient, sorted by age. ANOVA: F(59) = 9.629, *p* < 0.0001; post hoc: LSD). Each dot in B and C represents the calculation in one FOV of the tissue slice received from a patient. The numbers in parentheses indicate the 6–9 representative FOVs/JA tissue slice. Rc, recurrence; Rm, remission. (**D**) Examples illustrating the levels of LHCGR and CD271 expression in a sample from in a JA patient with JA recurrence. Relatively strong LHCGR immunoreactivity (*brown*) vs. much lower CD271 immunoreactivity (*red*) is evident near the small vessel lumina.

**Figure 6 cells-13-01217-f006:**
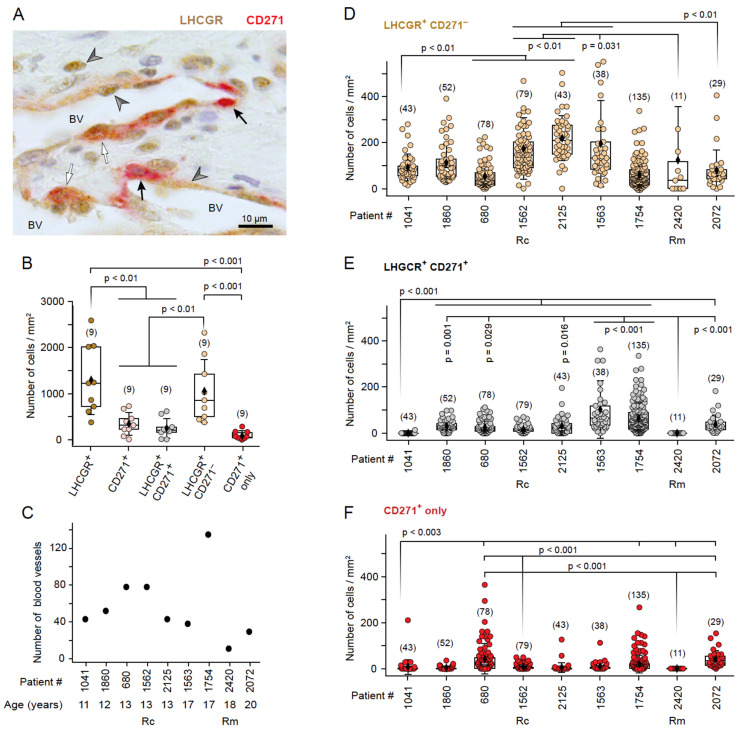
The JA vascular region contains a population of LHCGR^+^ CD271^+^ stem cells in close proximity to LHCGR^+^ CD271^−^ cells. (**A**) Immunohistochemical image showing LHCGR^+^ CD271^−^ cells (gray arrowhead), CD271^+^ cells (black arrow), and double-labeled LHCGR^+^ CD271^+^ cells (white arrow). BV, lumen of blood vessel. (**B**) Box plot of the number of cells/mm^2^ counted in the entire FOVs obtained from a sample of each patient (n = 9). ANOVA: F(40) = 11.389, *p* < 0.0001; post hoc: LSD. (**C**) Distribution of the number of blood vessels in each sample, sorted by age. (**D**) Box plots of the number of LHCGR^+^ CD271^−^ cells/mm^2^ near blood vessels as observed in samples from each individual patient, sorted by age. The numbers in parentheses indicate the total number of blood vessels analyzed for a given sample. Kruskal–Wallis ANOVA: χ^2^(8) = 185.45, *p* < 0.0001, post hoc: Dunn’s. (**E**) Box plots of the number of double-labeled LHCGR^+^ CD271^+^ cells/mm^2^ near blood vessels for a given sample of each individual patient, sorted by age. Kruskal–Wallis ANOVA: χ^2^(8) = 173.89, *p* < 0.0001, post hoc: Dunn’s. (**F**) Box plots of the number of CD271^+^ cells/mm^2^ near blood vessels containing no LHCGR for a given sample of each individual patient, sorted by age. Kruskal–Wallis ANOVA: χ^2^(8) = 121.59, *p* < 0.0001, post hoc: Dunn’s. Rc, recurrence; Rm, remission.

**Figure 7 cells-13-01217-f007:**
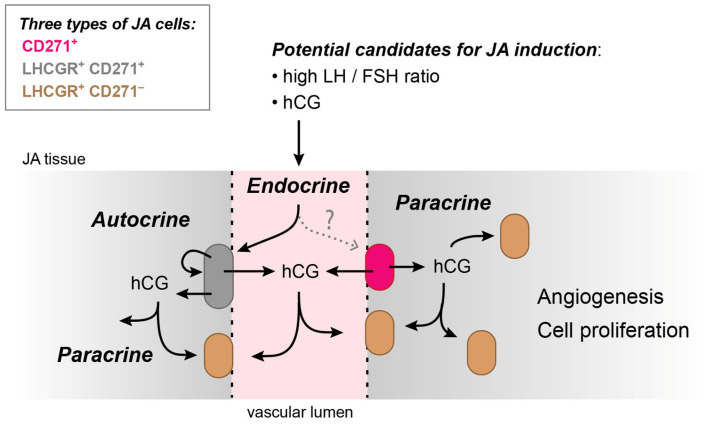
A model of JA induction based on the identification of three distinct JA cell types as observed in this study: CD271^+^ cells (*magenta*), double-labeled LHCGR^+^ CD271^+^ cells (*gray*), and LHCGR^+^ CD271^−^ cells (*brown*). An important implication is that positive feedback via endocrine, paracrine, and autocrine mechanisms may contribute to JA proliferation and the formation of new capillaries. Dashed arrow, unknown factor activating CD271^+^ cells.

## Data Availability

All data needed to evaluate the conclusions in the paper are present in the paper. Data (last edited on 18 July 2024) are deposited in Figshare (doi:10.6084/m9.figshare.25847089).

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
