# Peer review of "Widespread Distribution of Luteinizing Hormone/Choriogonadotropin Receptor in Human Juvenile Angiofibroma: Implications for a Sex-Specific Nasal Tumor"

_cells, 2024, doi:10.3390/cells13141217_

Round 1

Reviewer 1 Report

Comments and Suggestions for Authors

 In this study, the authors examined the cells expressing LHCGR and CD271 in human juvenile angiofibroma, and suggested the importance of these cells in the initiation/ progression of this tumor. The paper is basically well written and the results are interesting for the opening of new field in this tumor.

 The reviewer has minor comments.

Comments:

1, Is the anti-LHCGR molecule or anti CD271 molecule a possible inhibitor against this tumor ?

2, For the high vasculature phenotype of this tumor, which signaling system is an inducer of angiogenesis? Is VEGF system or others involved in this process?

Author Response

Point-by-point response to the comments of the reviewers

We were pleased with the overall positive, constructive, and enthusiastic reviews by the three Reviewers. We are grateful that they took the time to provide constructive comments, which enabled us to produce a better manuscript (ms). We believe that we have fully addressed their comments, through inclusion of editing the text at numerous places and providing a new supplemental figure (Figure S1). We now also extended the previous Figure 1, thus providing now the data of all five patients, as requested. To avoid loss of detail in the figures due to the addition of the extra patients, we have divided the original Figure 1 into a new Figure 1 and Figure 2. We have made numerous edits and improvements - they are explained in great detail in the point-by-point response. The edits in the ms are shown in red to facilitate comparison. All of these new results strengthen our point of view and improve the ms, without affecting our previous conclusions. We very much hope that after the substantial revision and expansion, our revised ms can be accepted for publication.

Reviewer 1:

 In this study, the authors examined the cells expressing LHCGR and CD271 in human juvenile angiofibroma and suggested the importance of these cells in the initiation/ progression of this tumor. The paper is basically well written and the results are interesting for the opening of new field in this tumor. The reviewer has minor comments.

The reviewer posed two inquiries: first, regarding the authors' opinion on the treatment of this tumor, and second, regarding the signaling system that triggers angiogenesis.

1, Is the anti-LHCGR molecule or anti CD271 molecule a possible inhibitor against this tumor ?

We agree with the reviewer that an antibody therapy might be useful in the treatment of JA. The new immune-based therapies are an effective tool for treating cancer, especially in advanced and metastatic stages (Izadpanah et al 2023 Cancer Medicine, PMID: 37698048). JA is not a type of cancer, but a tumor. The recurrence of this tumor is due to the persistence of residual tissue in the surgical area or at sites that cannot be resected due to their location and not metastasis. However, the administration of an anti-LHCGR molecule could interfere with the typical developmental processes of an adolescent JA patient, as the receptor plays an essential role in sexual maturation during puberty. It can be reasonably argued that this kind of systemic approach would be considered negligent.

An alternative approach could be the local treatment with anti-LHCGR injected into blood vessel supplying the angiofibroma prior to the cauterization of the main blood vessels. Cauterization would prevent the antibody from spreading throughout the patient’s body, thereby providing at least temporary relief from the recurrent nosebleed. Subsequently, the tumor size would require observation to ascertain whether a decrease in size is occurring over time, analogous to a spontaneous remission. Similarly, an anti-CD271 molecule could be combined with cauterization of the blood vessels. This has not yet been tested in human, as far as these authors are aware of. It has been investigated in cell culture resulting in a smaller tumor size (Morita et al. 2019 Cancer Letters, PMID: 31325530). Both suggestions are intriguing but would necessitate the initiation of a distinct clinical trial with ethical oversight. At this juncture, it is prudent to adopt a more conservative approach, whereby the hypothesis is initially substantiated by examining the hormonal serum levels of LH and hCG, as has been stated by us in the Conclusions (MDPI cells template: line 513 - 527).

2, For the high vasculature phenotype of this tumor, which signaling system is an inducer of angiogenesis? Is VEGF system or others involved in this process?

This is a very valid point, and while the exact mechanism by which angiogenesis is triggered in JA is still not clear, further research may help to shed light on this issue. The most frequently studied molecular markers are steroid receptors (androgen, estrogen, and progesterone receptors), followed by the vascular endothelial growth factor (VEGF). Although trends in the expression of these markers in JA were identified, no definitive conclusions could be drawn. As the reviewer correctly hinted to, VEGF is a known angiogenic factor and is indeed linked to JA (Liu et al., 2015 PMID: 25384380; Hota et al., 2015 PMID: 25890396). VEGF is not directly related to pubertal status and is largely not sex-specific, rendering it an inadequate explanation for the occurrence of JA in primarily adolescent males. It is however noteworthy that VEGF can be regulated by hCG and could promote angiogenesis via VEGF-MEK/ERK signaling (Jing et al., 2021 PMID: 33068962; Brouillet et al., 2012 PMID: 22138749). Consequently, the presence of LHCGR in JA tissue is intriguing, as hCG could potentially act as the trigger for angiogenesis in this benign sex-specific tumor. We have added this information into the manuscript in the

Introduction (MDPI cells template: line 81 - 90)

'Vascular endothelial growth factor (VEGF) is most commonly associated with an angiogenic effect, and is ubiquitously present in JA [35,36]. The relationship between VEGF and sexual dimorphism remains uncertain, and there is no clear evidence that it is related to pubertal status. Therefore, it is an inadequate explanation for the occurrence of JA in primarily adolescent males. hCG can however increase VEGF protein expression in a dose- and time-dependent manner [37] which, in turn, could promote angiogenesis via a MEK (mitogen-activated extracellular signal-regulated kinase) / ERK (extracellular signal-regulated kinase) signaling pathway [38]. Therefore, the activation of LHCGR could act as a trigger for angiogenesis in this benign sex-specific tumor.'

and Discussion (MDPI cells template: line 466 - 478):

'LHCGR has recently been considered as a prognostic marker for a subset of tumors, including the growth of testicular germ cell tumors (seminomas) [48]. Furthermore, LHCGR activity has been implicated in the induction and progression of some cancers [25,28]. Patients with a malignant endometrial adenocarcinoma have been found to benefit from treatments with inhibitors of either LHCGR or the vascular endothelial growth factor receptor 2 (VEGFR2) [49]. VEGF is a well-known angiogenic factor and a primary stimulant of tumor vascularization and it is widely present in JA [35,36]. VEGF is not directly related to pubertal status and is largely not sex-specific, rendering it an inadequate explanation for the occurrence of JA in primarily adolescent males. Importantly, hCG can upregulate VEGF protein expression in a dose- and time-dependent manner [37] which, in turn, could promote angiogenesis via a MEK/ERK signaling pathway [38]. hCG could thus potentially act as a trigger for angiogenesis in this benign sex-specific tumor.'

Reviewer 2 Report

Comments and Suggestions for Authors

I read and analyzed the manuscript "Widespread Distribution of Luteinizing Hormone/Chorio-gonadotropin Receptor in Human Juvenile Angiofibroma: Implications for a Sex-Specific Nasal Tumor" with great interest. The manuscript investigates the expression and localization of LHCGR in juvenile angiofibroma (JA) tissues. The topic is very interesting and timely and provides important new insights into understanding the hormonal regulation of JA that could potentially pave the way for targeted therapies.

Although the research provides new data on the expression of LHCGR in JA and adds a significant dimension to the understanding of hormonal influence in the pathogenesis of JA, a comparison with the existing literature on LHCGR in other vascular or hormone-dependent tumors is lacking to determine the broader relevance and novelty. Comparison with relevant research is recommended to further emphasize the significance of the findings.

The major criticism of this study is the sample size, although the methodological approach is sound as RNAscope and immunohistochemistry are used to accurately localize LHCGR. I suggest increasing the sample size where possible to increase the credibility of the results.

The statistical analysis is sound, but the manuscript would benefit from a more detailed description of the controls and validation steps to ensure the specificity and sensitivity of the RNAscope and immunohistochemistry results.

The authors hypothesize that LHCGR mediates JA vascularization and cell proliferation via endocrine, autocrine and paracrine mechanisms. Further functional analyzes are required to demonstrate the causal role of LHCGR in these processes.

 In the discussion, the results should be better linked to the existing knowledge of hormonal regulation in other similar tumors to provide a comprehensive understanding of the role of LHCGR in JA.

The manuscript is generally well written but contains areas where clarity could be improved. For example, an introduction could better define the gaps in current knowledge that this study attempts to address.

The figures are informative but would benefit from higher resolution and more detailed legends to aid understanding without extensive references to the main text. It is particularly important that the figures are placed immediately after the paragraphs in which they are mentioned so that the reader can immediately visualize the results described. In the first figure, it would be useful to show all samples from five patients to give a comprehensive overview of the variation in LHCGR expression. This would allow a better visualization and comparison between different patients.

Considering the rarity and clinical significance of JA, this research may provide valuable insights into the pathogenesis and management of this disease. Although the manuscript provides valuable insights into LHCGR expression in JA, revisions are needed to address the above concerns.

Author Response

Point-by-point response to the comments of the reviewers

We were pleased with the overall positive, constructive, and enthusiastic reviews by the three Reviewers. We are grateful that they took the time to provide constructive comments, which enabled us to produce a better manuscript (ms). We believe that we have fully addressed their comments, through inclusion of editing the text at numerous places and providing a new supplemental figure (Figure S1). We now also extended the previous Figure 1, thus providing now the data of all five patients, as requested. To avoid loss of detail in the figures due to the addition of the extra patients, we have divided the original Figure 1 into a new Figure 1 and Figure 2. We have made numerous edits and improvements - they are explained in great detail in the point-by-point response. The edits in the ms are shown in red to facilitate comparison. All of these new results strengthen our point of view and improve the ms, without affecting our previous conclusions. We very much hope that after the substantial revision and expansion, our revised ms can be accepted for publication.

Reviewer 2:

I read and analyzed the manuscript "Widespread Distribution of Luteinizing Hormone/Chorio-gonadotropin Receptor in Human Juvenile Angiofibroma: Implications for a Sex-Specific Nasal Tumor" with great interest. The manuscript investigates the expression and localization of LHCGR in juvenile angiofibroma (JA) tissues. The topic is very interesting and timely and provides important new insights into understanding the hormonal regulation of JA that could potentially pave the way for targeted therapies.

Although the research provides new data on the expression of LHCGR in JA and adds a significant dimension to the understanding of hormonal influence in the pathogenesis of JA, a comparison with the existing literature on LHCGR in other vascular or hormone-dependent tumors is lacking to determine the broader relevance and novelty. Comparison with relevant research is recommended to further emphasize the significance of the findings.

We agree with the reviewer and have now added a new paragraph to the Discussion linking LHCGR to the angiogenic factor VEGF, a well-recognized cancer biomarker and a primary stimulant of the vascularization of solid tumors. In this paragraph some additional literature is also cited (MDPI cells template: line 466 - 478):

'LHCGR has recently been considered as a prognostic marker for a subset of tumors, including the growth of testicular germ cell tumors (seminomas) [48]. Furthermore, LHCGR activity has been implicated in the induction and progression of some cancers [25,28]. Patients with a malignant endometrial adenocarcinoma have been found to benefit from treatments with inhibitors of either LHCGR or the vascular endothelial growth factor receptor 2 (VEGFR2) [49]. VEGF is a well-known angiogenic factor and a primary stimulant of tumor vascularization and it is widely present in JA [35,36]. However, VEGF is not directly related to pubertal status and is largely not sex-specific, rendering it an inadequate explanation for the occurrence of JA in primarily adolescent males. Importantly, hCG can upregulate VEGF protein expression in a dose- and time-dependent manner [37] which, in turn, could promote angiogenesis via a MEK/ERK signaling pathway [38]. hCG could thus potentially act as a trigger for angiogenesis in this benign sex-specific tumor.'

Furthermore, a comparison was made to existing literature on LHCGR in relation to other vascular or hormone-dependent physiology, diseases and tumors. These citations are part of other references as detailed below. For example, we indicated in the Introduction (3rd paragraph):

 '... hCG not only promotes the proliferation of stem cells but can also be secreted by abnormal germ cells in benign or malignant tumors [20-24].' (MDPI cells template: line 68 - 70)

and in the Discussion (1st paragraph) that 'hCG, which is important during pregnancy and plays a role in various tumors [24,32,33].' (MDPI cells template: line 430 - 431).

Furthermore, in the Discussion, we cite evidence indicating that LHCGR plays a role in promoting new capillary development in ovarian endothelial cells [34,47] (MDPI cells template: line 463 - 465)

and cite literature on elevated hCG levels in female benign or malignant tumors (20-24) (MDPI cells template: line 501 - 503) and male malignant tumors [25,26,52] (MDPI cells template: line 503 - 507).

The major criticism of this study is the sample size, although the methodological approach is sound as RNAscope and immunohistochemistry are used to accurately localize LHCGR. I suggest increasing the sample size where possible to increase the credibility of the results.

The statistical analysis is sound, but the manuscript would benefit from a more detailed description of the controls and validation steps to ensure the specificity and sensitivity of the RNAscope and immunohistochemistry results.

First of all, we would like to thank the reviewer for stating that our statistical analyses are sound. Nevertheless, the reviewer raises concerns about the sample size. In response to this concern, we would like to point out that the sample size, although of limited number because of reasons stated below, is fully sufficient to support the major results and conclusions of this novel report. Otherwise, our statistical analyses would not have been sound. We were particularly careful in this ms to focus on the most solid and convincing findings and not to overstretch our interpretations of the results. A total of 11 patients were included in the study, with tissue from five JA patients used for RNAscope, nine JA patients used for immunohistochemistry, and one ovarian cancer patient used for both RNAscope and immunohistochemistry. The size of the JA tissue samples varied considerably, rendering them unsuitable for use in both techniques. We have been unable to (1) identify additional JA patients and (2) patients willing to provide their tissue for research purposes. One major challenge here is that JA is a rare tumor with an incidence of only 1:5,000-1:50,000 in otolaryngology patients (Gullane et al 1992 Laryngoscope PMID 1323003). Although it is always better to increase sample size ( see ORPHA# 289596) this is currently not possible for us and, in our opinion, is not necessary for our conclusions. We have carefully pointed out the limitations of this first report and indicated a logical series of follow-up experiments for the years the come (MDPI cells template: line 512 - 527). A such, we believe that our study provides an important contribution to a field which has made relatively little progress over the past years.

We agree with the reviewer that the manuscript would benefit from a more detailed description of the controls and validation steps to ensure specificity and sensitivity. In paragraph 2.2. and 2.4. (MDPI cells template: lines 122 - 128 and lines 181 - 183), we now added the following:

'Negative and positive RNAscope controls were performed (Figure S1). As negative control, we used a channel 1 probe for the dapB gene of Bacillus subtilis strain SMY provided by the manufacturer (Cat#320871, ACD Bio-Techne). As positive control, we used ovarian tissue obtained from an ovarian carcinoma patient (#2346-1) that expresses LHCGR. This tissue was kindly provided by Prof. Dr. R. M. Bohle, Institute of Pathology, University Hospital of Saarland, Germany.',

and

'As a negative control, we applied antibody dilution buffer without primary antibodies (Figure S1). As positive control, we used ovarian tissue obtained from an ovarian carcinoma patient (patient #2346-1) which expresses LHCGR and CD271 (Figure S1), kindly provided by Prof. Dr. R. M. Bohle, Institute of Pathology, Saarland University Medical Center.'

We also provided a new supplemental Figure S1 containing the controls of the RNAscope and immunohistochemistry. The legend contains additional descriptive information:

'Supplementary Material: Figure S1. Validation of RNAscope and immunohistochemistry experiments. (A) RNAscope positive control for LHCGR on an ovarian tissue section from an ovarian carcinoma patient (#2346-1). LHCGR+ cells (red) are visible. DAPI (4′,6-diamidino-2-phenylindole, blue) indicates the location of nuclei. (B) RNAscope negative control reaction using a tissue section from JA patient #1754. The RNAscope negative control was a channel 1 probe for the dapB gene of Bacillus subtilis strain SMY provided by the manufacturer (Cat#320871, ACD Bio-Techne). No specific staining was detected. DAPI indicates the location of nuclei. RNAscope controls of the examples depicted in A and B were performed on the same day and with the same solutions except for the used probe. LHCGR RNAscope labeling from the same patient is depicted in Figure 2J - L. (C) As a positive control for the immunohistochemical experiments, an ovarian tissue section from the same patient (#2346-1) as in A was used. LHCGR+ (brown), CD271+ (red). (D) Negative control from the same tissue as in C. Both LHCGR and CD271 primary antibodies were omitted resulting in no signal for these probes. The tissue sections of C and D were performed on the same day and with the same solutions except for the presence of the primary antibodies. (E) Negative control in a tissue section from JA patient #1754 in which both LHCGR and CD271 primary antibodies were omitted resulting in no signal. A positive control from the same patient using anti-LHCGR is depicted in Figure 4G. (C - E) Nuclei are stained using hematoxylin.'

The authors hypothesize that LHCGR mediates JA vascularization and cell proliferation via endocrine, autocrine and paracrine mechanisms. Further functional analyzes are required to demonstrate the causal role of LHCGR in these processes.

The reviewer points out that ‘further functional analyses are required to demonstrate a causal role of LHCGR….’. Although this is certainly true, these experiments (using human tissue) would take years and would go far beyond the scope of this initial report. With all due respect and given the editorial mandate to provide a revised version of the paper within 10 days, these experiments cannot be part of this first study and need to be performed in future work.

In the discussion, the results should be better linked to the existing knowledge of hormonal regulation in other similar tumors to provide a comprehensive understanding of the role of LHCGR in JA.

As the reviewer requested at the beginning of his review, we have not only extended a paragraph in the Introduction (MDPI cells template: line 81 - 90), but now also added a new paragraph to the Discussion section (MDPI cells template: line 446 - 478), regarding LHCGR in other hormone-dependent tumors.

The manuscript is generally well written but contains areas where clarity could be improved. For example, an introduction could better define the gaps in current knowledge that this study attempts to address.

We agree with the reviewer that more clarity was necessary to link LHCGR to the main player of angiogenesis in cancer and tumors, VEGF. We have thus extended a paragraph in the Introduction (MDPI cells template: line 81 - 90) that refer to the relationship between LHCGR and VEGF in the context of vascularization and sex/age specificity. This is particularly relevant in elucidating the underlying mechanisms of JA in predominantly male adolescents:

'Vascular endothelial growth factor (VEGF) is most commonly associated with an angiogenic effect, and is ubiquitously present in JA [35,36]. The relationship between VEGF and sexual dimorphism remains uncertain, and there is no clear evidence that it is related to pubertal status. Therefore, it is an inadequate explanation for the occurrence of JA in primarily adolescent males. hCG can however increase VEGF protein expression in a dose- and time-dependent manner [37] which, in turn, could promote angiogenesis via a MEK (mitogen-activated extracellular signal-regulated kinase) / ERK (extracellular signal-regulated kinase) signaling pathway [38]. Therefore, the activation of LHCGR could act as a trigger for angiogenesis in this benign sex-specific tumor.'.

In addition to the candidate, LHGGR, which links sex-specificity to trigger angiogenesis, the Introduction also highlighted several other knowledge gaps related to JA. These included the cause of sex specificity, the reason for its effect on a specific age group, and the manner in which LHCGR may act via its multiple ligands in physiological processes both during development and in tumors:

(1) Introduction, 1st paragraph (MDPI cells template: line 44 - 49): '....investigations of sex hormone receptors and hormone blood levels for androgen, estrogen and progesterone led to contradictory results [7,9-11] so that no general acceptance of a hormonal cause could be established. As an alternative receptor candidate, the luteinizing hormone/choriogonadotropin receptor (LHCGR) was previously identified in JA tissue by RT-PCR, but this did not attract much attention since then [10].'

(2) Introduction, 2nd paragraph (MDPI cells template: line 50 - 67): The paragraph examines the impact of LHCGR, which acts as a ligand for multiple physiological agonists (LH and the five different hCG ligands), on physiological processes during development.

(3) Introduction 3rd paragraph (MDPI cells template: line 68 - 90): The paragraph examines the impact of LHCGR and specific hCG ligands on physiological processes in tumors. This paragraph has now been extended to include the new citations regarding VEGF which are very useful.

The figures are informative but would benefit from higher resolution and more detailed legends to aid understanding without extensive references to the main text. It is particularly important that the figures are placed immediately after the paragraphs in which they are mentioned so that the reader can immediately visualize the results described. In the first figure, it would be useful to show all samples from five patients to give a comprehensive overview of the variation in LHCGR expression. This would allow a better visualization and comparison between different patients.

We are grateful to the reviewer for this request, as our original intention was to provide an overview of all patients and their individual variability. In order to avoid the loss of details in the figures due to the addition of the extra patients, we have divided the original Figure 1 into a new Figure 1 and Figure 2. Samples from all five patients are now shown to give a comprehensive overview of the variation in LHCGR expression. The original figures have a resolution of 1200 dpi and were uploaded separately to the MDPI cells server. However, the images in the MDPI cells Word template had been reduced in size to allow easy access to the text. Otherwise, the manuscript file size would have been enormous. Thus the problem of perceived low resolution should not occur in the final publication.

In accordance with the reviewer's request, each legend has been extended and now provides a brief title. This indeed facilitates the comprehension of the figure content for the reader.

The proposed placement of the figures was intended to be in close proximity to the text that references them. Due to the size of some figures, it is not always possible to place them immediately after the paragraph in which they are mentioned. In some instances, the figures are incorporated within the aforementioned paragraph or immediately preceding it. Ultimately, we believe that the Editors/Publishers will have a final say regarding the precise placement of the figures at their optimal positions.

Considering the rarity and clinical significance of JA, this research may provide valuable insights into the pathogenesis and management of this disease. Although the manuscript provides valuable insights into LHCGR expression in JA, revisions are needed to address the above concerns.

In summary, we have added significant new data and performed a great number of edits to improve our work. We hope the reviewer will be satisfied with these revisions.

Reviewer 3 Report

Comments and Suggestions for Authors

Several studies investigated the link existing between hormonal imbalances and tumor development and/or progression (see for example PMID: 38892035). In this contex, previous work reported that the luteinizing hormone/choriogonadotropin hormone receptor (LHCGR) is expressed in juvenile angiofibroma (JA), a benign but highly vascularized and rapidly growing nasal tumor that is mostly prevalent in male adolescents [PMID: 23996526].

Based on this finding, the present study evaluated, via the use of RNAscope and immunohistochemistry, the expression and localization of LHCGR in JA tissue samples. Results indicated that most of LHCGR-positive cells also expressed the stem cell marker CD271 and were localized in the vascularized areas of the JA, in close proximity to the vessels lumen. This finding suggests that LHCGR-positive cells are susceptible to endocrine signaling, and provides an explanation for the gender specificity and pubertal manifestation of JA.

In my opinion, the present study is original and relevant for the field as it adds useful information to the narrow literature published on this specific topic. Importantly, the study highlights new diagnostic/ prognostic markers, as well as new therapeutic targets, for JA. The employed methodology and the quality of the presented data, tables and figures are very good. However, regarding the conclusions and the cited references, Authors should better clarify, and further stress, that JA is a bening tumor, although it is locally aggressive and it frequently relapses after surgical resection and/ or resists to radiotherapy [PMID: 31623023; PMID: 26019389]. In particular, referring to, and discussing about, topics such as metastasis should be drastically minimized.

Author Response

Point-by-point response to the comments of the reviewers

We were pleased with the overall positive, constructive, and enthusiastic reviews by the three Reviewers. We are grateful that they took the time to provide constructive comments, which enabled us to produce a better manuscript (ms). We believe that we have fully addressed their comments, through inclusion of editing the text at numerous places and providing a new supplemental figure (Figure S1). We now also extended the previous Figure 1, thus providing now the data of all five patients, as requested. To avoid loss of detail in the figures due to the addition of the extra patients, we have divided the original Figure 1 into a new Figure 1 and Figure 2. We have made numerous edits and improvements - they are explained in great detail in the point-by-point response. The edits in the ms are shown in red to facilitate comparison. All of these new results strengthen our point of view and improve the ms, without affecting our previous conclusions. We very much hope that after the substantial revision and expansion, our revised ms can be accepted for publication.

Reviewer 3:

Several studies investigated the link existing between hormonal imbalances and tumor development and/or progression (see for example PMID: 38892035). In this contex, previous work reported that the luteinizing hormone/choriogonadotropin hormone receptor (LHCGR) is expressed in juvenile angiofibroma (JA), a benign but highly vascularized and rapidly growing nasal tumor that is mostly prevalent in male adolescents [PMID: 23996526].

Based on this finding, the present study evaluated, via the use of RNAscope and immunohistochemistry, the expression and localization of LHCGR in JA tissue samples. Results indicated that most of LHCGR-positive cells also expressed the stem cell marker CD271 and were localized in the vascularized areas of the JA, in close proximity to the vessels lumen. This finding suggests that LHCGR-positive cells are susceptible to endocrine signaling, and provides an explanation for the gender specificity and pubertal manifestation of JA. 

In my opinion, the present study is original and relevant for the field as it adds useful information to the narrow literature published on this specific topic. Importantly, the study highlights new diagnostic/ prognostic markers, as well as new therapeutic targets, for JA. The employed methodology and the quality of the presented data, tables and figures are very good. However, regarding the conclusions and the cited references, Authors should better clarify, and further stress, that JA is a bening tumor, although it is locally aggressive and it frequently relapses after surgical resection and/ or resists to radiotherapy [PMID: 31623023; PMID: 26019389]. In particular, referring to, and discussing about, topics such as metastasis should be drastically minimized.

We agree with the reviewer that the fact that JA is a benign tumor should be clearly stated in the manuscipt, as we did several times throughout the ms, for example already in the second sentence of the Introduction. In principle, the term benign tumor expresses clinically that there is no metastasis. The term “metastatic” has been utilized only on one occasion in the manuscript, specifically in connection with citation to the literature. However, based on particular inquiries from the other reviewers, we believe that the term "recurrence" may have given the impression of metastatic involvement which is indeed incorrect for JA.

As the reviewer correctly points out, JA often relapses. This recurrence of JA is not due to metastasis of cells. This would have changed the classification of JA to a malignant tumor. The problem is the persistence of residual tissue in the surgical area or at sites that cannot be resected due to their location. For anatomical reasons and in most cases, it is simply not possible to perform a complete resection on JA with the required safety margin. The vascular changes observed in JA and, in light of this study, the CD271 stem cells and the LHCGR cells near the vessels are difficult to spot during resection without markers. Therefore, recurrence of JA can be defined as the persistence of residual JA tissue.

Instead of repeating multiple times that JA is a benign tumor, we opted to clarify the probable cause of the misunderstanding and the probable reason of JA recurrence. Several sentences have been added at the first mention of JA recurrence (1st paragraph in 3.2 of the Results section) to make this pont fully clear:

' The recurrence of this benign tumor is assumed to be due to residual tumor tissue that could not be removed during surgery because of its location.'  (MDPI cells template: line 299 - 301);

and at the end of that same paragraph

'Given the anatomical constraints of surgical resection, complete removal of JA may fail. Consequently, recurrence represents here a prevalent phenomenon which can be defined as the persistence of residual JA tissue. Stem cells are the reactive cells that trigger cell proliferation, regardless of whether the nasal cavity of healthy patients contains LHCGR+ cells. The recurrence of JA may be attributed to the presence of CD271+ cells in proximity to small vessels, which could render LHCGR+ cells susceptible to endocrine stimulation, thereby contributing to the recurrence of JA.'. (MDPI cells template: line 355 - 340 and 355 - 356).

We hope that the reviewer will be satisfied with these changes and explanations.